

# Sustaining Low-Cost PM2.5 Monitoring Networks in South Asia: Technical Challenges and Solutions

Talha Saeed[1], Muhammad Mahad Khaliq[2], Michael Howard Bergin[3], Prakash V. Bhave[3], Noora Khaleel[4], Enna Mool[5], Mahesh Senarathna[6], Shahid Uz Zaman[7], Shatabdi Roy[8], Abdus Salam[8], Jas Raj Subba[9], Muhammad Fahim Khokhar[1]*

5    1 Institute of Environmental Sciences & Engineering (IESE), School of Civil & Environmental Engineering (SCEE), National University of Science & Technology (NUST), Islamabad – 44000, Pakistan

2 School of Electrical Engineering & Computer Sciences (SEECS), National University of Science & Technology (NUST), Islamabad – 44000, Pakistan

3 Department of Civil and Environmental Engineering, Duke University, Durham, NC 27708, USA

4 Department of Environment and Natural Science, Faculty of Engineering Science and Technology, The Maldives National University, Male' 20371, Maldives

10   5 Central Department of Environmental Science, Tribhuvan University, Kirtipur, Kathmandu, Nepal

6 Postgraduate Institute of Science, University of Peradeniya, Peradeniya – 20400, Srilanka

7 Department of Chemistry, Faculty of Science, Bangladesh University of Engineering and Technology, Dhaka-1000, Bangladesh

8 Department of Chemistry, Faculty of Science, University of Dhaka, Dhaka-1000, Bangladesh

Department of Physical Science, Sherubtse College, Kanglung Bhutan

*Correspondence to*: Muhammad Fahim Khokhar (fahim.khokhar@iese.nust.edu.pk)

**Abstract.** The need to monitor South Asia's air quality stems from its significant negative effects on human and environmental health. Traditional, regulatory-grade air quality monitoring systems have proven costly to operate and very difficult to maintain in most South Asian countries. Low-cost sensor (LCS) networks have been touted as a viable alternative, but the challenges to sustain them have not been evaluated or thoroughly documented. the acceptance of such monitors, in particular by regulatory

agencies, across South Asian countries is still lacking. Lack of acceptance is due to prevailing myths (especially, in the regulatory circles of South Asia) about their accuracy, precision, consistency, dependability, maintenance, and calibration concerns. The present study attempts to fill that knowledge gap while also providing practical solutions to enhance the longevity of LCS, perhaps adding years to their lives. Specifically, this study describes strategies and maintenance plans for operating large networks of TSI BlueSky (8143) Sensors across South Asia, with a focus on problems caused by power outages,

power surges, weather conditions, and continued exposure to high amounts of dust and pollution. The article provides further support that incorporating LCS networks into the regulatory framework can facilitate the enforcement of environmental regulations and legislation against polluters. The goal is to develop a more reliable and long-lasting air quality monitoring system that will assist South Asian countries to reduce air pollution-related health hazards and consequent socio-economic disruptions.





**Graphical Abstract**

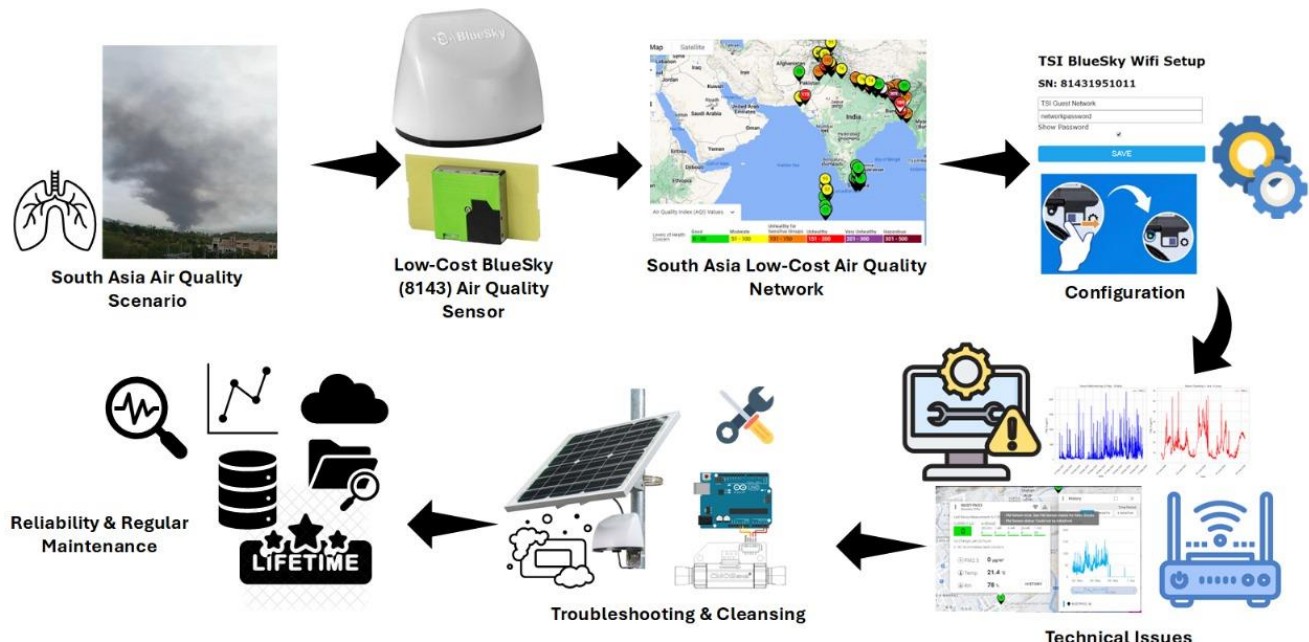

Map Credits: © Google Maps 2024, and TSI Link, 2024

**Keywords**: PM$_{2.5}$, BlueSky, South Asia, malfunctioning, solutions

**Highlights**

- LCS network is a realistic solution for South Asian air quality monitoring.
- LCS-related problems can be fixed with routine maintenance and debugging.
- The lifetime of LCS can be extended by many years with timely diagnostics and maintenance.





## Introduction

### 1.1 Air Pollution & its Effects

Six criteria pollutants are mostly responsible for atmospheric pollution which include ozone ($O_3$), particulate matter (PM), carbon monoxide (CO), lead (Pb), sulfur dioxide ($SO_2$) and nitrogen dioxide ($NO_2$) (Saxena & Sonwani, 2019). Chatterjee et.al., (2023) concluded that in 2019, 1.02 million deaths in South Asia were attributed to PM emissions from fossil fuel

burning and industrial processes (Anjum et al., 2021). PM is a condensed mixture of dust and hygroscopic particles in the environment, which are further categorized by size as coarse particles ($PM_c$) and fine particles ($PM_{2.5}$) coming from natural and anthropogenic sources (Griffin, 2013). Natural sources can be dust storms, forest fires & volcanic eruptions (USEPA, 2022). Human activities like biomass burning, and vehicular and industrial emissions are the major contributors to atmospheric $PM_{2.5}$ (Daellenbach et al., 2020). According to the World Health Organization, 99% of the world's population is breathing air

unhealthy levels of $PM_{2.5}$ (Javellana, 2022). Not only does the PM explicitly pollute the environment, but also has malignant health impacts on human beings (Goswami & Neog, 2023). Fine particulates can enter respiratory airways, deposit in the lungs and penetrate into the bloodstream (Hussain et al., 2011). In addition, PM affects the earth's radiative balance thereby contributing to global climate change (Wang et al., 2023).

### 1.2 Air Quality of South Asia and Monitoring Challenges

The air quality recorded in South Asian regions reached the most hazardous levels in recent years. During very high pollution episodes, especially in winter, some cities like Lahore have closed schools and limited vehicle movement to mitigate human exposures (Khokhar et al., 2021; Khokhar et al., 2023). Currently, many countries in the region cannot solve their air quality problems due to economic or political instability, lack of infrastructure and government will (Anjum et al., 2021).

Pakistan, Bangladesh, Sri Lanka, Nepal, Bhutan and the Maldives have all established air quality monitoring networks at

various times in the past 25 years, but the vast majority have been short-lived (Bilal et al., 2021; Dhammapala et al., 2022; Prakash et al., 2021). In most cases, initial capital costs have been budgeted internally or provided by development agencies to purchase and install regulatory-grade air quality monitors like the MetOne Beta Attenuation Monitor (BAM) and GRIMM Environmental Dust Monitor (EDM) (Majeed et al., 2024; Mehadi et al., 2020). After a few enthusiastic years of government- or NGO-led monitoring, the long-term operation and maintenance are often outsourced to the private sector. When numerous

unexpected expenses later arise, funds dry up and the networks cease to function (Din et al., 2023; Abdul Jabbar et al., 2022). Despite their rigorous calibration and dependability results, developing countries need help accessing these accurate instruments for an expansive portable air quality monitoring network. (Borrego et al., 2018). With the invention of micro-controllers, the manufacturers developed sensors for monitoring $PM_{2.5}$ which are inexpensive, portable, and easy to install as they require only a minimal watt of electricity connection and a Wi-Fi connection (Kuncoro et al., 2022).



### 1.3 Low-cost Particulate Matter Sensors

Low-cost sensors (LCS) are a relatively new terminology introduced to South Asian nations where air pollution is a major concern for the region because of increasing events like smog, heat waves and floods (Lung et al., 2022). LCS network is one of the only viable options for the region due to the provision of remote data access and little maintenance compared to reference monitors (Feenstra et al., 2019; Zikova et al., 2017). The credibility of LCS is often questionable because of their accuracy and bias (Huang et al., 2022). Sensor sensitivity is susceptible to unique artifacts caused by electronic faults or other abnormalities (Chen et al., 2021). To correct the biases of LCS, prolonged collocation and calibration with reference monitors are needed (Kelly et al., 2017; Datta et al., 2020). LCS are of various types depending on the PM component used inside the sensor (Zamora et al., 2020). The sensor performance and evaluation in various studies emphasized only calibration, statistical analysis, spatial or temporal validation and machine learning modelling (Johnson et al., 2018; Zheng et al., 2018; Zheng et al., 2019; Liu et al., 2019; Madhwal et al., 2024; Bi et al., 2020). No study has been conducted in which the maintenance of a vast LCS network across the region and the challenges faced by the South Asia countries are discussed, to troubleshoot for the long-term sustainability of an air quality monitoring network.

### 1.4 TSI BlueSky (8143) Air Quality Monitor

TSI BlueSky (8143) sensor is comprised of a Sensirion SP30 pm component, micro-controller, and SD card, and can measure $PM_{2.5}$, $PM_{10}$, Temperature & Relative Humidity (BlueSky air quality monitor 8143, 2021). The Sensirion SPS30 particulate matter sensor provides an optical sensing mechanism to quantify airborne particulate matter. Its operation is centered around the employment of a source of light e.g. a laser or an LED to shine on particles as they flow within the sensor's chamber resulting in the detection of the scattered light by a photodetector that is strategically placed. It serves as the foundation for the sensor's capability of differentiating particle size and concentration with the help of complex signal processing algorithms. The SPS30 comprises an internal fan that induces a continuous airflow that results in enhanced accuracy and reliability of readings. This characteristic along with the robust design of the sensor and factory calibration provides stability and negligible maintenance requirements over time (Sensirion, 2021).

### 1.5 Challenges towards acceptability of Low-cost particulate monitor sensors

With the advent of cheap PM sensors, the debate regarding environmental regulation has been ignited because these sensors can take monitoring to a wider public, produce more detailed data, and thus weaken the monopoly level of centralized institutions (Keyes et al., 2023). Low-cost sensors are an instance of an affordable as well as portable way of measuring air quality which has never been found to be feasible through expensive monitoring stations (Farooqui et al., 2023; Raysoni et al., 2023). This does not mean that these cannot be used, but they are accepted if the criteria of accuracy, reliability and consistency of data output are strictly met (Genikomsakis et al., 2018). Regulatory authorities require a validation requirement to be met to overcome these constraints (Danek & Zaręba, 2021). Sensors should be designed so that they can work effectively even in





hostile environments. Sensor data should be calibrated and validated using standard procedures discussed in the literature (Zheng et al., 2018) to determine whether the sensors can be utilized in coordination with environmental policy and regulatory measures (Badura et al., 2018). These novel technologies hold great promise for improving air quality and health monitoring, but a thorough examination of their accuracy, precision, and degradation over time is required before incorporating them into

traditional air quality management (Kunugi et al., 2018).

### 1.6 South Asia Capacity Building Project

Under the support of the U.S. Department of State, Duke University initiated a project to build the capacity of South Asian countries to improve air quality through data-driven policy decisions (Thompson, 2020). One of the main objectives was to install low-cost sensors across the South Asia region to monitor air pollution and the other was to build the capacity of these

countries for acceptability of the low-cost sensors network data into the government policymaking and regulations (Building capacity to improve air quality in South Asia, 2021). To debunk the myth that low-cost sensors have less lifetime, reliability, and acceptability, this study aims to highlight the robustness of BlueSky $PM_{2.5}$ Monitor and strategies to improve their lifetime. Moreover, the study provides the solutions to the challenges faced by having the LCS network across the region and capacity building to overcome the constraints that might help local regulatory agencies to opt for LCS networks for Air Quality

management and devising adequate mitigation strategies.

For this study, the TSI BlueSky (8143) Sensor network of 380 sensors was deployed across South Asia. The sensors were installed in countries including Pakistan, Sri Lanka, Nepal, Maldives, Bangladesh, and Bhutan (Table 1).

**Table 1: TSI BlueSky (8143) Sensors Country-wise Count**

| Country | Pakistan | Sri Lanka | Nepal | Maldives | Bangladesh | Bhutan |
|---|---|---|---|---|---|---|
| **Deployment** | December 2021 | July 2021 | September 2021 | September 2021 | July 2021 | September 2021 |
| **BlueSky $PM_{2.5}$ Sensors Count** | 61 | 57 | 71 | 20 | 75 | 20 |

All the sensors needed to be registered on the TSI Link website (TSI Link Solutions, 2023), and then harmonized by collocating

the sensors all together in one place for a specific period, averages were compared to the median values, and the correction factors were calculated, and applied to each sensor' settings option available on TSI Link dashboard (Figure 1).







**Figure 1: TSI BlueSky 8143 Air Quality Sensors Network of South Asia (Grey color signs indicate sensors shut down due to electrical or internet issues in the region) [Image Credit: TSI Link, 2024 Map Credit: © Google Maps, 2024]**

### 1.6.1. Installation & Configuration

The sensors were configured using the IP Address (192.168.4.1) by entering the Wi-Fi username and password in the settings. A shift of the button towards the operating position activates the sensor. The sensor indicates its status through the blinking of a green light: one blink per second signifies a successful internet connection and data transmission to the cloud, two blinks indicate an incorrect Wi-Fi password, and three blinks suggest an inability to transmit data to the cloud, likely due to the





absence of an internet connection. Additionally, successful connection status can be verified through the TSI Link Dashboard, which displays all registered sensors associated with a specific account. The sensor features a dome shape with a silver mesh for air intake and exhaust adjacent to the micro-USB power supply port and configuration button (Figure 2).

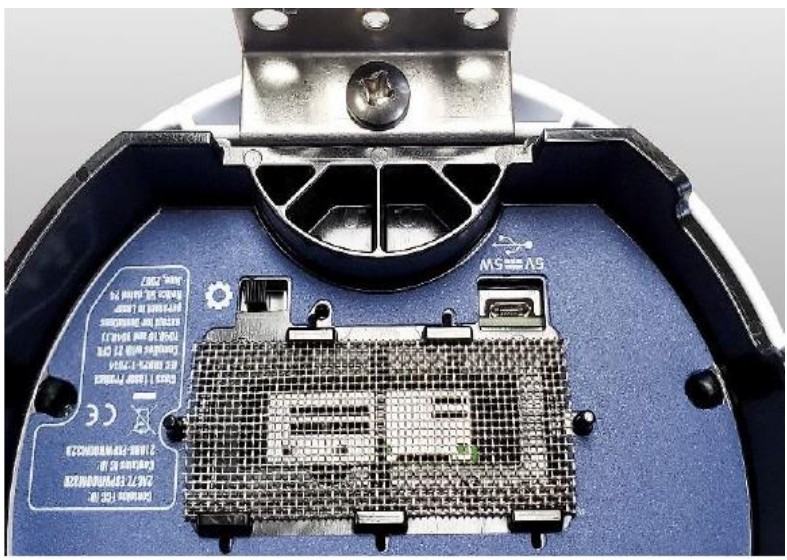

**Figure 2: Underneath section of dome-shaped BlueSky 8143 Sensor  (TSI Bluesky Operational Manual)**

**1.6.2. Data Downloading**

Country representatives created accounts on TSI Link, registering devices with their serial numbers, names, locations, and calibration factors. The platform also allows for adjusting settings, sharing, relocating, and visualizing historical data.

There are three ways to get data from BlueSky devices.

1)      Running TSI-provided scripts to download Quality Control (QC) and Quality assured (QA) data in CSV format

2)      Download data from the TSI dashboard (providing three months of data without QA & QC)

3)      Manual extraction from the SD card (without QA & QC)

An annual subscription is required for API access and data downloading via the first and second methods.



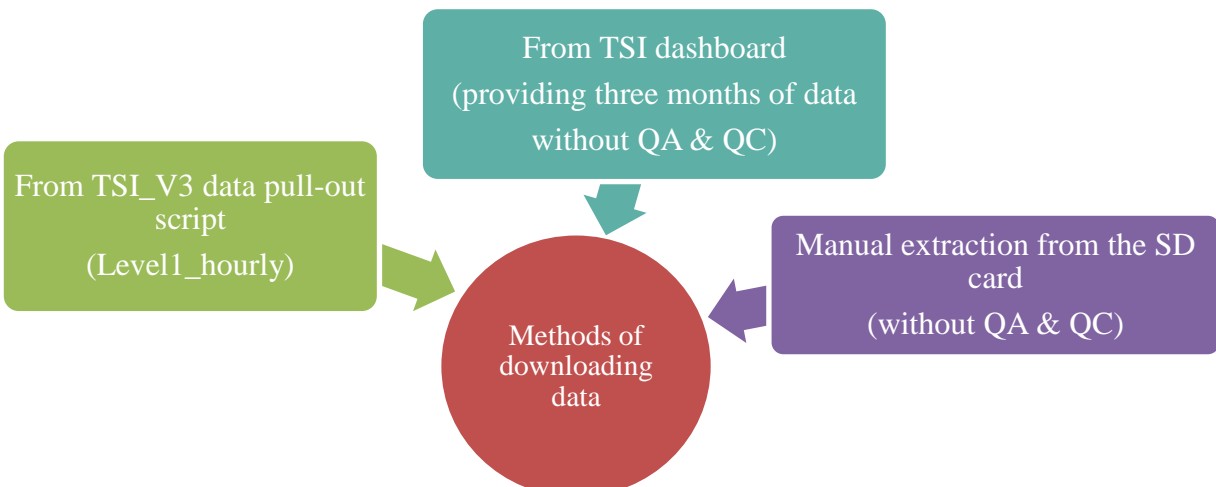

### 1.6.3. Sensor Harmonization

The data collected from each sensor was compared to the median value of the total number of co-located sensors. A linear regression analysis was conducted on each sensor to get the best line of fit that indicates the relationship between its measurements and the median value. A linear regression with zero intercepts was determined to be appropriate, and the multiplicative harmonization factor for each sensor was derived using the equation below:

$$\text{harmonization factor} = \frac{1}{\text{slope of the linear regression with zero intercept}}$$

### 2. Methodology

This section of our research paper outlines the comprehensive methodology adopted to troubleshoot, debug, and fix BlueSky air quality monitoring nodes. These nodes, deployed across diverse environmental conditions in South Asia encountered issues such as electricity disruptions, halting data transmission, and reporting zero or anomalously high $PM_{2.5}$ values. The following subsections detail the steps taken to address these challenges, ensuring the reliability and accuracy of our air quality monitoring network:

### 2.1 Symptoms and Initial Diagnostics

The Sensirion SPS30 particulate matter sensors, known for their precision in monitoring air quality, have a recommended operational lifetime of 1 year under specified conditions as per TSI Blusky Manual. Despite their robust design, premature failures were observed, attributed primarily to dust accumulation and exposure to extreme environmental conditions. Deployed nodes exhibited symptoms such as complete interruption of data output, zero readings, or implausibly high $PM_{2.5}$ values (Figure



3). These issues prompted a systematic approach to troubleshooting, beginning with the node's removal from its field location and disconnection of the sensor from the PCB board for initial diagnostics.

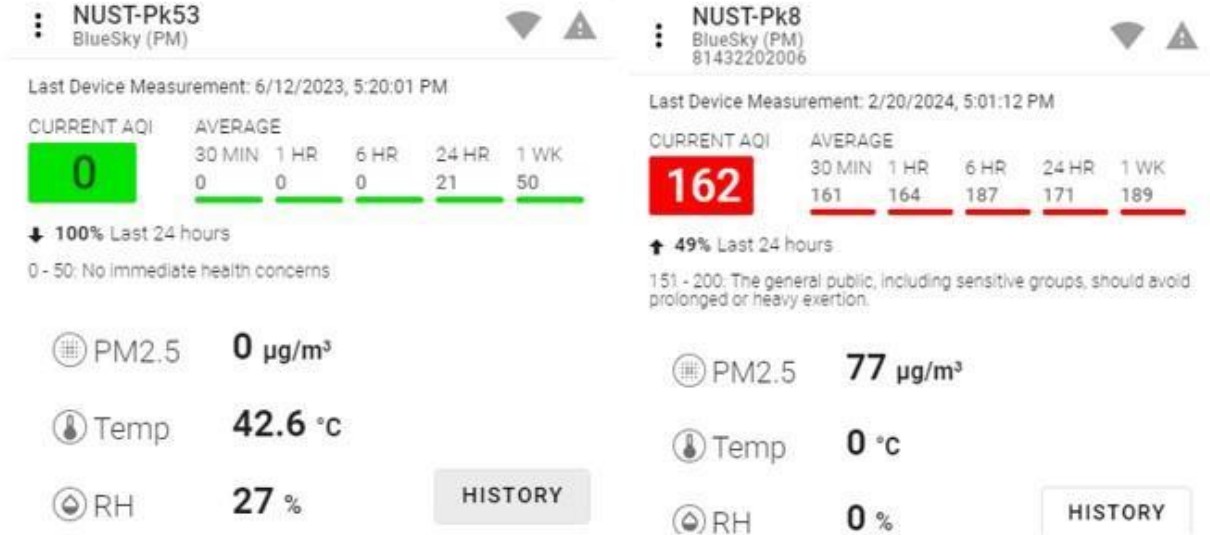

**Figure 3:** TSI Dashboard information showing Zero Values for PM2.5 (left) and Zero Values for Temperature & Humidity (Right) Image Credit: TSI Link Dashboard www.tsilink.com

## 2.2 Removal of Dust Accumulation

Following the initial diagnostics, the sensors were gently unmounted and inspected physically for any apparent damage. Almost every sensor would require cleaning. The cleaning technique involved gentle opening of the case to check for dust
obstructions that can restrict the airflow route, preventing the sensor laser and fan from functioning properly.

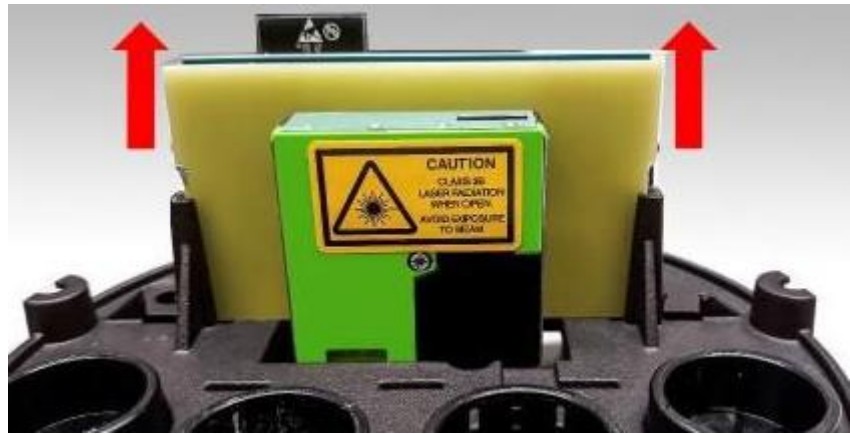

**Figure 4:** Sensirion PM Component disassembly procedure for cleansing purposes (Image Credit: TSI Bluesky Manual)





Air is first blown into the vent of the fan, followed by disassembling the sensor module using the locks and screws. Once the
sensor is disassembled, we can see the intricate components residing inside it. Dust particles are visible most of the time. By
using a fine brush and air blower, the dust is blown out of the channel. The identified halting dust deposits clogging the interior
airflow channel were handled with extreme caution. This cleaning step was crucial for restoring the sensors' ability to precisely
measure particulate matter. After this, the installed fan is also cleaned.

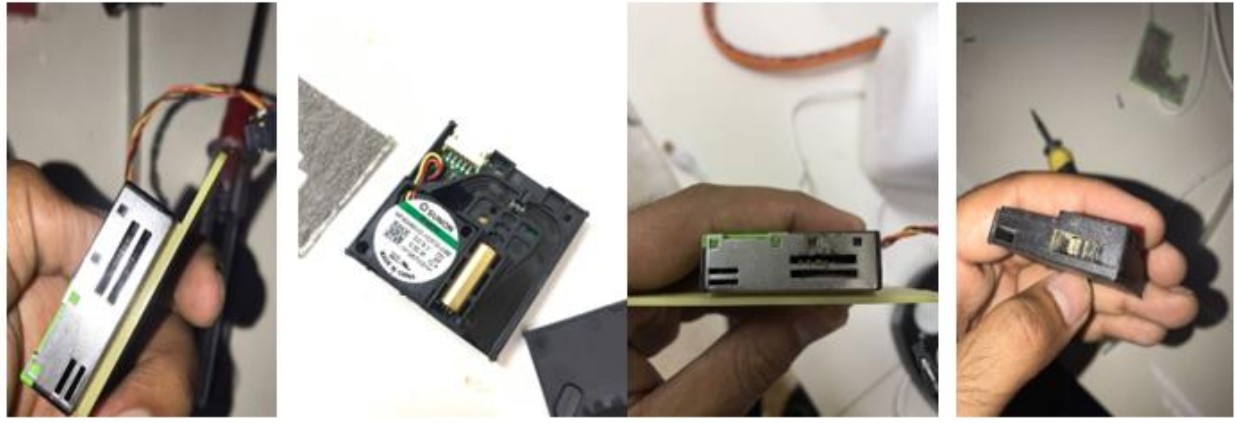


**Figure 5:** Cleansing mechanism of PM Sensor component of BlueSky 8143 Air Quality Monitor

### 2.3 Troubleshooting and Debugging

Once thoroughly cleaned, the sensor units were reassembled and subjected to a 'cleansing mode' operation via the Arduino
platform. This step was essential to verify the restoration of the sensor's performance to its expected operational standards.
Despite their robust design, premature failures were observed, attributed primarily to dust accumulation and exposure to
extreme environmental conditions. Deployed nodes exhibited symptoms such as complete interruption of data output, zero
readings, or implausibly high PM2.5 values. Due to these challenges, debugging had to be performed systematically. To begin,
the sensor had to be removed from its field location and unplugged from the PCB board to perform more diagnostics.

### 2.4 Sensor Connectivity and Functionality Tests

The first step involved interfacing the removed SPS30 sensor with microcontrollers, specifically Arduino Uno or ESP32, to
conduct initial functionality tests. Under the functionality tests include power input to the sensor, connector of sensor cable,
and sensor's internals including circuitry, laser module, small air chamber. Initially the input power to the sensor is diagnosed,
if there is voltage across the terminals, the input power is being transferred to the sensor unit, if the digital multimeter fails to
do so, the sensor is completely out of order and there might be a possibility of loose connector which is to be addressed. Once
the input power is checked, the sensor is to be inspected for the connector that connects it with the microcontroller. Due to
humid environment, sometimes the terminals of the connector are rusted, which are to be cleaned. At this stage, the connector



is also checked for loose contact. Following the sensor's connector, the internals of the module are to be disassembled and checked thoroughly. The important modules of the internals comprise of a laser unit, air chamber, dc fan, printed circuit board and a filter. The cleansing process at this part accounts for solving the problem of very high or 0 values from the sensor. If

operated under extreme dust conditions, dust accumulates at the fan and the air chamber and sometimes it is also present in abundance inside the air flow path. The fan is taken out gently and first air is blown at the wings of it, then at the shaft, dusting off particles stuck over it. Once air is blown off the fan, using isopropyl alcohol and cotton buds it is rinsed off gently. After this air is blown at the air chamber and laser unit, following cleansing using isopropyl alcohol. These tests were executed in both UART and I2C communication modes to ascertain the sensor's operational status, determining whether it was non-

responsive or producing inaccurate readings (Figure 6).

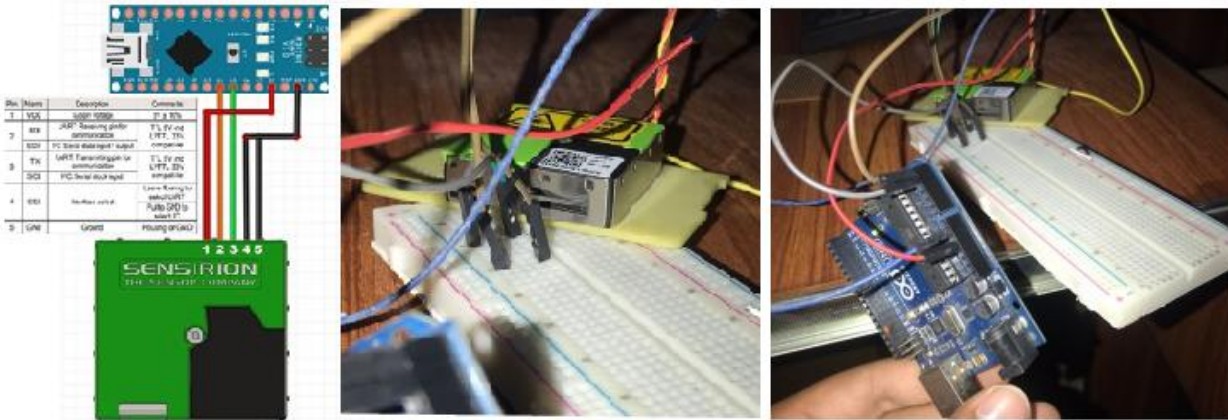

**Figure 6:** Sensor Debugging (Sensirion Connection Image Credit: https://www.instructables.com/How-To-Sensirion-SPS30/) This detailed troubleshooting and repair methodology has enabled us to significantly enhance the reliability and accuracy of the BlueSky air quality monitoring network across Pakistan. By systematically addressing the challenges posed by

environmental conditions and sensor wear, we have ensured that our network continues to provide vital data for assessing air quality and protecting public health.

### 3. Results & Discussion

### 3.1 South Asian Countries Feedback

Table 3 below illustrates the issues faced by the countries and the solutions they adopted to overcome the encountered

problems:

**Table 2:** Summary of the technical challenges and solutions in maintaining BlueSky sensor network in South Asian countries.

| Country | Issues Faced | Proposed Solutions / Trouble shootings |
|---|---|---|
| Sri Lanka | • Power supply issues, | 1. Installing dedicated power outlets and opting solar powered backups, |





| | | |
|---|---|---|
| | • Network connectivity, <br> • Sensor placement, <br> • Calibration, <br> • Maintenance, <br> • firmware updates <br> • Malfunctioning sensors | 2. using Wi-Fi dongles, <br> 3. following manufacturer instructions for sensor placement, <br> 4. regular calibration and maintenance, <br> 5. manual firmware updates |
| Nepal | • Power supply disruptions, <br> • Wi-Fi connectivity (Frequent change in Wi-Fi password), <br> • Monitor defects. <br> • Firmware updates | 1. Coordination with local partners, physical and virtual training, <br> 2. Opting for the solar-powered backup, <br> 3. Downloading data from SD card, <br> 4. Replacing malfunction adapters |
| Maldives | • Permission for sensor deployment, <br> • Continuous Wi-Fi and power supply, <br> • Lack of reference stations for calibration, <br> • sensor maintenance due to dust and salinity in the air | 1. Raising awareness about air pollution issues and the efficacy of monitoring <br> 2. Using external modems, downloading data from SD cards and securing power switches <br> 3. Calibration against regional reference stations <br> 4. Cleaning sensors regularly |
| Bhutan | • Wi-Fi connectivity, <br> • firmware upgrades, <br> • data recording intervals, <br> • remote area access and road obstructions | 1. Verifying internet and sensor connections, <br> 2. exploring firmware upgrade solutions, <br> 3. efficient logistics for remote access, collaborating with local authorities for road clearance |
| Bangladesh | • Wi-Fi access (Frequent change in Wi-Fi password), <br> • Electricity disruptions, <br> • Sensor maintenance, <br> • Sensor defects <br> • Security breaches <br> • Firmware updates | 1. Using cellular modems, <br> 2. Replacing malfunctioning adapters <br> 3. Regular cleaning, <br> 4. Regular physical inspection and constant communication with hosts <br> 5. Involving the public and ensuring security concerns while selecting the areas/locations to deploy the sensors |
| Pakistan | • Bad Electronic Board | 1. Replacement with locally assembled board |





| | | | |
|---|---|---|---|
| | ● Sensirion Component Malfunctioning | 2. | Cleansing & replacement of component if rainwater goes inside the sensirion. |
| | ● Dead Power Supplies | 3. | Locally available same specifications power adapter with little modifications |
| | ● Webs & Bugs inside the sensor compartment | 4. | Monthly cleaning, maintenance, and check ups |
| | ● Internet Disruptions | 5. | Wi-Fi dongles for internet provision |
| | ● Harsh Weather conditions | 6. | Replaced of worn-out connector pins from local market |

## 3.2 Technical Faults & Troubleshooting

In the South Asian region, the deployment of low-cost environmental sensors encountered significant challenges primarily due
to electricity disruptions. The predominant issue was the failure of power adapters and electronic boards, attributed to the
region's prevalent high voltages and extreme heat conditions. The design of the power adapters, encapsulated in silicone to
enhance durability in harsh environments, unfortunately rendered them irreparable upon failure. Consequently, when adapters
failed, the only recourse was replacement with locally sourced equivalents that matched the original specifications. Similarly,
the electronic boards, once damaged, could not be repaired, necessitating either the procurement of replacements from the
original manufacturer or the adoption of localized alternatives. These localized electronic boards offered a distinct advantage
in data management, allowing for storage within a local network as opposed to reliance on cloud-based solutions.

To address internet connectivity challenges, particularly in remote areas, the implementation strategy included the use of Wi-
Fi SIM dongles equipped with USB-B type male-female y-splitters. This setup facilitated internet access through the
subscription to mobile data plans. However, the advent of 5G technology introduced compatibility issues, as the sensors were
not equipped for 5G connectivity. This necessitated a fallback to 4G networks to maintain sensor functionality and data
transmission integrity.



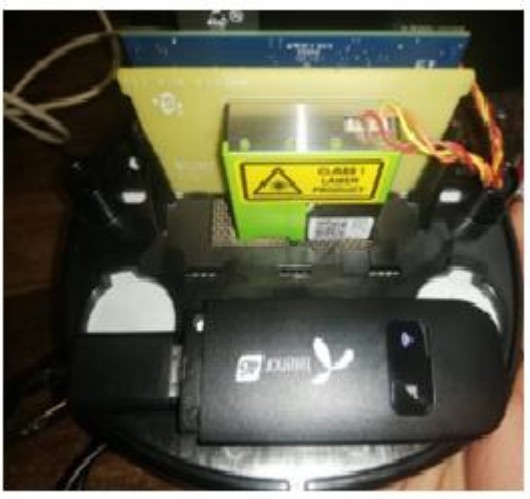 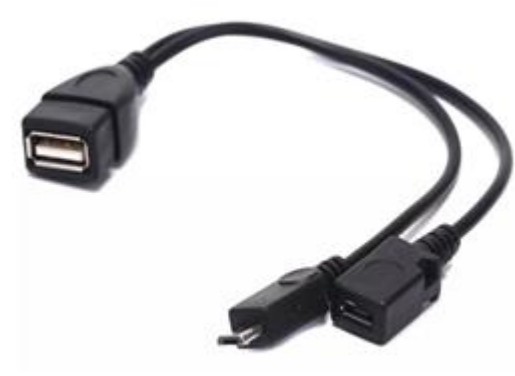

**Figure 7:** Wi-Fi Dongle Setup within the Air Quality Monitor Box using Y splitter connection cable.

The project also faced hurdles related to internet reliability across the region. In certain remote locales, internet access was
either unreliable or absent, exacerbating the risk of data loss. This was particularly problematic when sensors, unable to connect
to the 5G network or any internet service due to weak signals, failed to transmit data to the cloud or local storage solutions
such as SD cards.

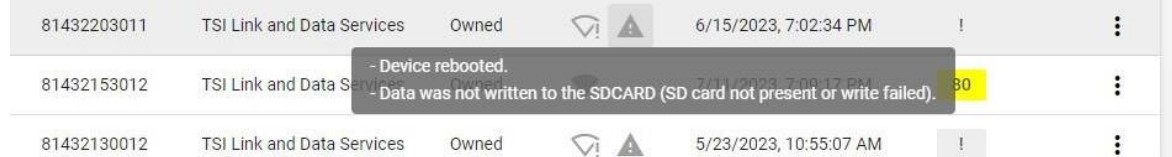

**Figure 8:** Internet-related Malfunctioning (Image Credit: TSI BlueSky Dashboard).

An additional challenge emerged from environmental pollutants, which compromised the functionality of particulate matter
(PM) sensors. The heavy pollution loads led to the clogging of sensor components, resulting in erroneous zero-value readings
that inaccurately suggested a clean atmosphere. To mitigate this, a thorough cleaning protocol was established, involving the
removal of dust from beneath the steel mesh and the disassembly of the sensor casing to directly cleanse the PM component
as mentioned in the methodology above.




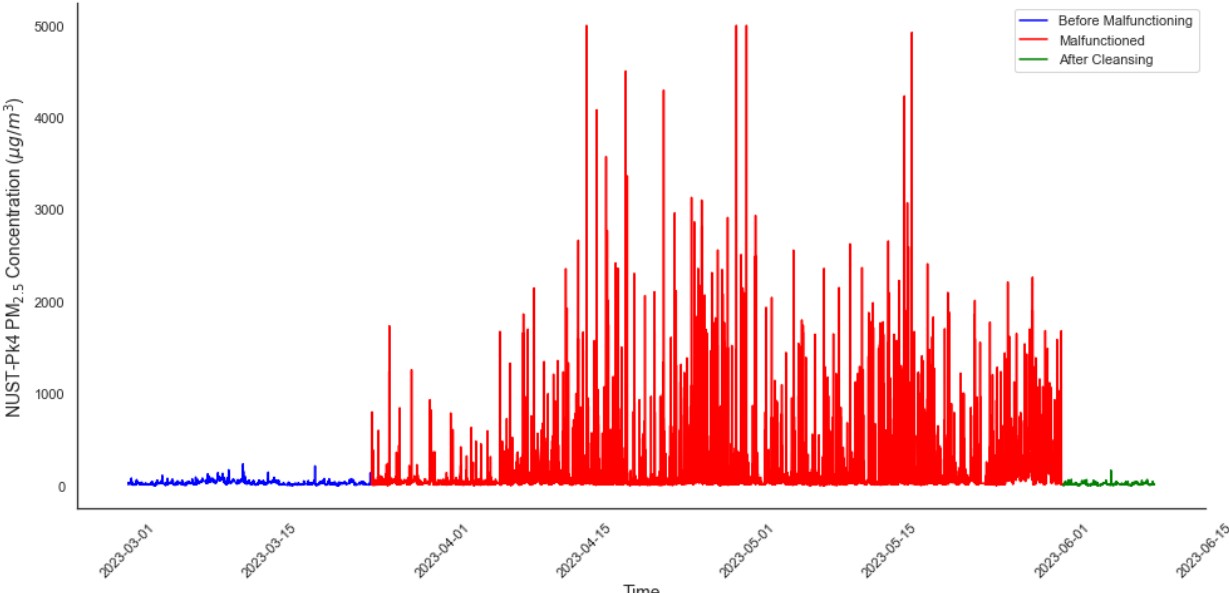

**Figure 9:** Time series depicting the before malfunction, malfunction and after cleansing results of TSI BlueSky PM$_{2.5}$ (µg/$m^3$) Monitor installed in Phalia, Punjab, Pakistan

The comparison was drawn to evaluate the correction factors variance for low-cost sensors both pre- and post-cleansing to
ascertain the effectiveness of the cleansing mechanism. The calibration process involved placing low-cost sensors alongside a BAM, which serves as a reference monitor, over a designated time interval. The readings from the sensor were then compared against those of the BAM to establish correction factors, which were calculated based on the median value of the discrepancies observed. After this initial calibration, the sensors underwent a cleansing process intended to remove any contaminants that may affect their performance. Following cleansing, the sensors were recalibrated to determine if the cleansing had any impact
on their accuracy. The lack of variance in correction factors pre- and post-cleansing suggests that the cleansing mechanism does not adversely affect the sensor's calibrated state. This finding is pivotal as it establishes the viability of the cleansing process as a maintenance procedure for these low-cost sensors. It also underscores the stability and reliability of the sensors' performance post-cleansing, an essential characteristic for long-term field deployment.

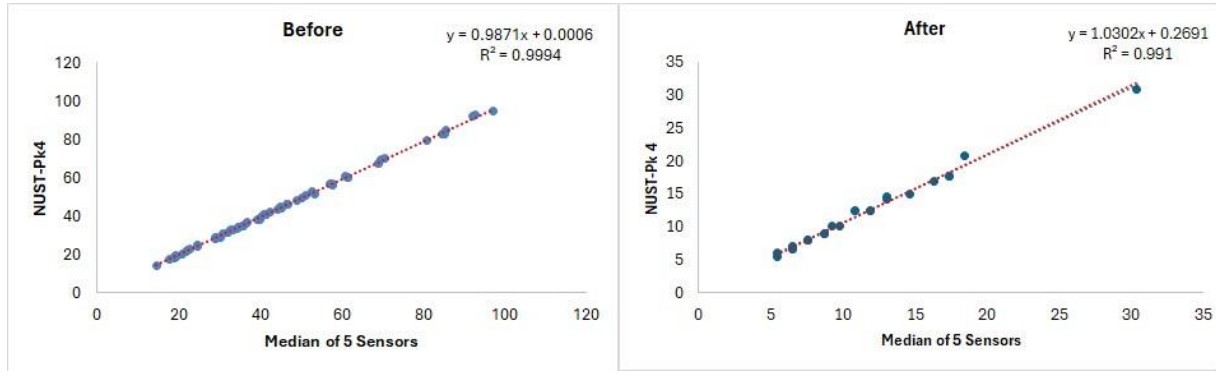





**Figure 10:** Air Quality Harmonization before (6-9Dec, 2021 )and after(18-19Apr, 2024) cleansing of NUST-Pk4.

**Table 3:** Sensor Harmonization factor before and after cleaning procedure

| Sr. No | Harmonization Date before cleansing | Sensor Tag | Before cleansing Correction Factor | Harmonization Date after cleansing | After cleansing Correction Factor |
|---|---|---|---|---|---|
| 1 | 22-25th Apr 2022 | NUST-Pk17 | 1.15 | 17-18th Apr 2024 | 1.32 |
| 2 | 10-16 May 2022 | NUST-Pk18 | 1.03 | 17-18th Apr 2024 | 1.05 |
| 3 | 20-27 May 2023 | NUST-Pk50 | 1 | 17-18th Apr 2024 | 0.93 |
| 4 | 27-31 May 2022 | NUST-Pk55 | 1.27 | 17-18th Apr 2024 | 0.72 |
| 5 | 15-29 Dec 2021 | NUST-Pk2 | 1.18 | 18-19th Apr 2024 | 1.12 |
| 6 | 15-29 Dec 2021 | NUST-Pk4 | 0.92 | 18-19th Apr 2024 | 0.95 |
| 7 | 15-29 Dec 2021 | NUST-Pk6 | 1.05 | 18-19th Apr 2024 | 1.01 |
| 8 | 10-16 May 2022 | NUST-Pk34 | 1.38 | 18-19th Apr 2024 | 1.38 |
| 9 | 17-20 May 2022 | NUST-Pk42 | 1.13 | 18-19th Apr 2024 | 1 |
| 10 | 15-29 Dec 2021 | NUST-Pk3 | 0.97 | 19-21st Apr 2024 | 0.84 |

## 3.3 Environmental Stressed Conditions

Air quality sensors are critical components in monitoring environmental pollution and providing data for public health advisories. However, these instruments are susceptible to environmental factors that can impair their functionality. In Islamabad, weather extremes, such as heat waves, and power outages present significant challenges to the operational integrity of these sensors. This part examines the correlation between harsh weather conditions, specifically heat waves, and the likelihood of sensor malfunctions due to rusting of pins, decreased power output, and power supply failures.






**Figure 11:** Meteoblue Islamabad Weather Date from 1ˢᵗ Jan 2023- 1ˢᵗ Jan 2024 (Weather archive Islamabad, 2024)

The temperature panel indicates that Islamabad experiences high temperatures, particularly from May to August, with peaks that could potentially exceed 40°C. Concurrently, relative humidity levels during these months show a decrease, which suggests the occurrence of heat waves. Such conditions are known to accelerate the corrosion process, likely leading to rusting of sensor pins.



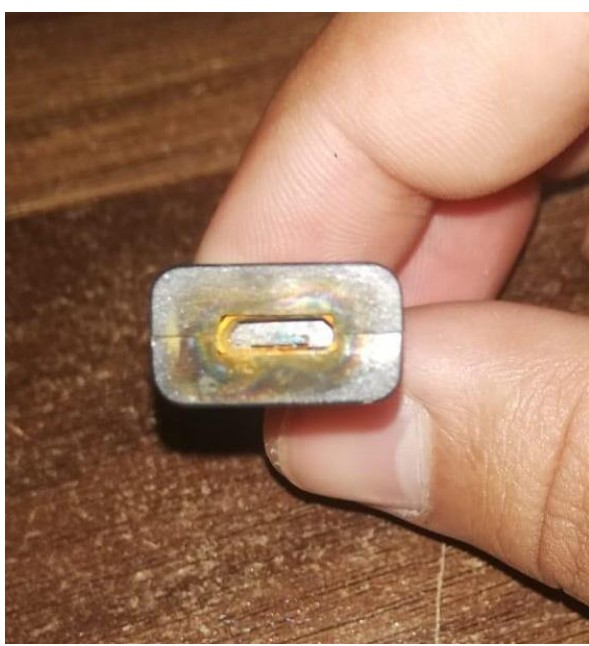

**Figure 12:** The worn-out rusty female USB B type pin of y splitter attached with Wi-Fi Dongle in Islamabad

The precipitation chart shows substantial rainfall in July and August. While precipitation itself might not directly cause rusting, the subsequent increase in humidity levels after rainfall can create an environment conducive to corrosion, particularly if sensors are not adequately protected. Wind can act as a catalyst for heat dissipation, which might help in cooling down the sensors during heat waves. However, it can also introduce particulate matter that could accumulate on sensors, potentially leading to power output reduction.

### 3.4 Power Outrages & Solar Powered sensor

In addressing the critical issue of power outages within the context of low-cost sensor network deployment, the adoption of solar-powered sensors emerges as a pivotal solution, contributing significantly to the sustainability and resilience of environmental monitoring efforts (Pandiyan et al., 2023). Regular monitoring intervals are essential in ensuring the consistent performance and reliability of these sensors, thereby mitigating the risks associated with intermittent power supply.

**Table 4:** Solar-powered Air Quality Monitor Power input and output results

| Items | Specifications | Solar Test | Trial 1 | Trial 2 |
|---|---|---|---|---|
| Solar Plate Max Watt | 30W | Start Date | 27/10/2022 19:00 | 15/11/2022 09:30 |
| Max. Voltage | 18V | Stop Date | 14/11/2022 09:30 | 18/11/2022 12:15 |
| Max. Power Current | 1.56A | Total Run Time | 422 hrs. | 74 hrs. |
| Battery | 312Wh (12V-26Ah) | | | |





The transition towards solar-powered solutions is not merely a technical adjustment but also aligns with broader sustainability objectives, particularly those encapsulated within the Sustainable Development Goals (SDGs). The integration of solar energy

in sensor networks reflects a commitment to sustainable energy practices (SDG 7) and responsible consumption and production (SDG 12), addressing the environmental concerns associated with electronic waste in developing regions (Falcone, 2023). The prevalent practice in third-world countries to repair rather than discard electronic components underscores a resourceful approach to managing electronic waste (Rauf, 2024). However, the limitations of repair options for certain components like sealed adapters necessitate the exploration of alternative solutions, such as solar power, which offers a sustainable and reliable

power source for sensor networks.

The validation of sensor performance, both before and after the cleansing mechanism, is crucial in establishing the efficacy and reliability of this transition. Validation processes ensure that the sensors maintain accurate and consistent data collection capabilities, despite the challenges posed by power outages (Liang, 2021). Moreover, the lifetime of these sensors can be significantly extended through proper maintenance practices, enhancing the overall sustainability and cost-effectiveness of the

sensor network deployment (Giordano et al., 2021). The adoption of solar-powered sensors in response to power outages presents a viable and sustainable solution that aligns with global sustainability goals. Through regular monitoring, rigorous validation, and diligent maintenance, the longevity and reliability of low-cost sensor networks can be significantly enhanced, contributing to the robustness of environmental monitoring initiatives in regions prone to power disruptions.

**3.5 South Asian Air Quality Sensor Uptime**

The sensor network not only provided these South Asian countries with an accurate air quality network but also had a high uptime. The discussion (Table 3) showed that the countries, especially at the beginning of the deployment stage, faced numerous challenges. However, after this phase, the uptime of the network averaged more than 50 percent ( Figure 13).



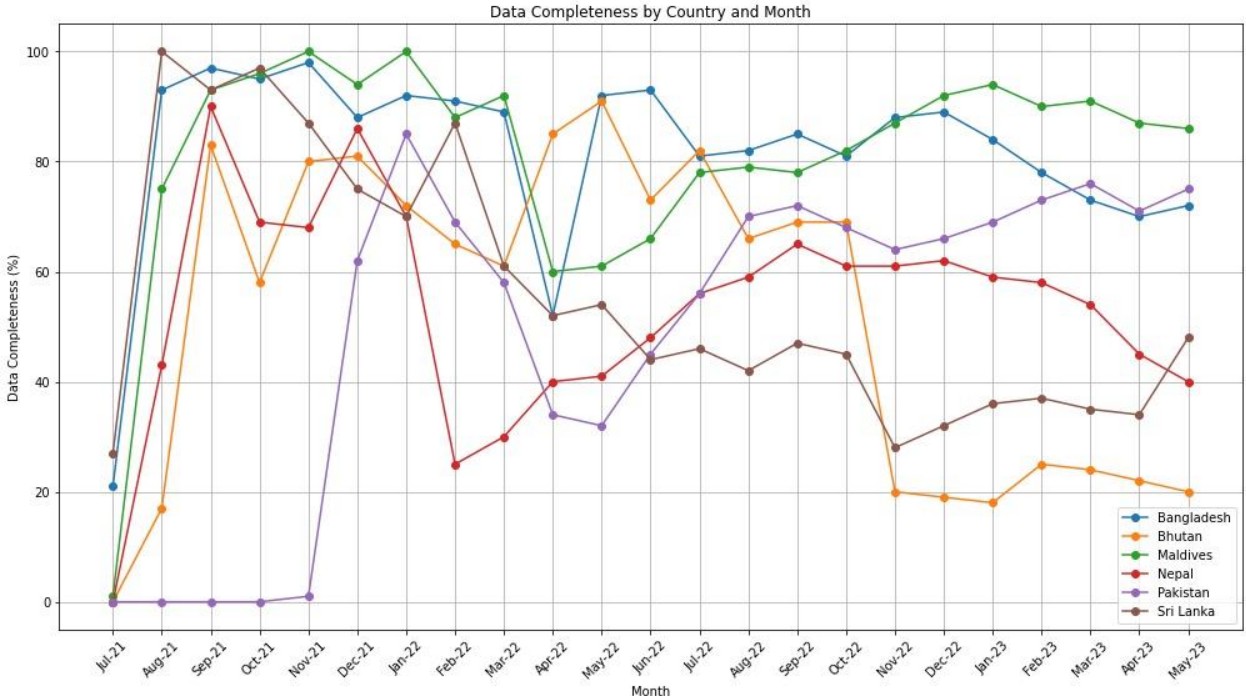

**Figure 13:**TSI BlueSky Air Quality Network Uptime in South Asian Countries

**Conclusions**

In conclusion, the acceptability of low-cost PM$_{2.5}$ sensors in South Asia remains limited, mainly due to capacity-building challenges among regulators and prevailing concerns regarding the sensors' longevity compared to reference monitors. However, as highlighted in the study, the implementation of strategic measures such as regular maintenance, calibration and

rigorous validation can significantly mitigate these limitations. By adhering to a regular maintenance schedule and conducting validations biannually, the operational lifespan of low-cost sensors can be extended to several years, thereby enhancing their viability and reliability in air quality monitoring. This approach not only addresses the concerns associated with durability but also enlightens the potential of low-cost sensors as a sustainable solution for air quality assessment in resource-constrained countries.

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
