# Peer review of "Sustaining Low-Cost PM2.5 Monitoring Networks in South Asia: Technical Challenges and Solutions"

_EGUsphere, 2024_

## Author Comment (AC1)

**Reply to the comments of Referee # 1**

**Preprint Details:**

**Manuscript number:** egusphere-2024-1932

**Title:** Sustaining Low-Cost PM2.5 Monitoring Networks in South Asia: Technical Challenges and Solutions

**Authors:** Talha Saeed, Muhammad Mahad Khaliq, Michael Howard Bergin, Prakash V. Bhave, Noora Khaleel, Enna Mool, Mahesh Senarathna, Shahid Uz Zaman, Shatabdi Roy, Abdus Salam, Jas Raj Subba, Muhammad Fahim Khokhar*

**Authors Acknowledgment:**

The authors sincerely thank the referees for their thorough reviews and insightful comments. We have carefully addressed each comment by acknowledging the constructive feedback and incorporating it into the revised manuscript to enhance its quality and clarity.

**Referee # 1 Comments:**

This manuscript provides an examination of the technical and logistical challenges in deploying and maintaining low-cost $PM_{2.5}$ sensor networks across South Asia (excluding India). It addresses region-specific issues such as power outages, environmental stressors, and maintenance demands, and offers strategies to mitigate these challenges. The work contributes insights into air quality monitoring in low-resource settings, with a focus on practical solutions for scaling low-cost sensors (LCS). However, there are fundamental concerns related to sensor calibration that would require more than analysis and writing adjustments to address.

**Specific comments:**

1. The authors assert that maintenance strategies can "debunk the myth" of limited sensor lifespan. However, broader literature indicates that while proactive maintenance can improve sensor performance, low-cost sensors still face durability constraints, particularly in high-dust, high-temperature, and humid environments. Addressing these limitations more directly would help set realistic expectations for LCS durability. Additionally, the lack of discussion on calibration raises questions about the study's claims regarding sensor lifespan.

**Reply:** The paper addressed the limitation by presenting the results of low-cost sensors installed at locations with heavy pollution load and extremely hot temperature in South-Asian countries. Regular maintenance includes calibration of the sensor in a calibration chamber every 1 year after following cleaning and maintenance protocols. The paper is meant to serve as a maintenance protocol for low-cost sensors as the low-cost sensors manufacturing company claims only 1 year for the sensor lifetime, but the network is sustained for more than 3 years in several South-Asian countries due to consistent maintenance protocols on a regular basis. For instance, Figure 10 depicts one sensor in Pakistan (NUST-Pk4) which was exposed to severe dust episode caused by construction activity in the proximity of it (after cleaning NRMSE=0.033), and it has been successfully restored by following rigorous quality control, maintenance and calibration procedure as mentioned in section 2 (Methodology). Calibration and collocation are described in other companion manuscripts Jain et al (2024); Madhwal et al. (2024); Khaleel et al. (2024); Shabbir et al. (in review); Shrestha et al. (in review); Zaman et al. (in preparation); Senarathna et al. (in review).

2. While the study is presented as covering South Asia, it does not include India—the region's largest country. Providing a rationale for excluding India would be valuable for readers, helping clarify the study's scope. Referring to this work as a "South Asia-wide study" might create an impression of broader coverage than is provided, so revising this framing would improve accuracy.

**Reply:** This is a very good point. Although India was part of the overall project funded by the US Department of State (PI M. Bergin) due to geo-political issues with Pakistan, the Indian team could unfortunately not participate in this publication. It is worthwhile to point out that the Indian team did roll out a network of ~70 BlueSky sensors in urban Lucknow that collected data over a 3-year period that is described by Madhwal et al. 2024 (Atmos. Environ.) and Jain et al. 2024 (Atmos. Environ.).  The context is still focused on South-Asian countries having Nepal, Maldives, Sri Lanka, Bhutan, Bangladesh, and Pakistan that in essence surround India and represent the broad range of meteorology and particulate matter sources and concentrations experienced in India. With this said we have replaced "South Asia" with "South Asian countries" in the title.

3. Although the manuscript details maintenance protocols, it lacks information on the costs associated with these activities. A cost-benefit analysis comparing the maintenance of LCS with that of traditional regulatory monitors would offer policymakers clearer insights into the economic feasibility of implementing LCS networks at scale.

**Reply:** The cost of a low-cost sensor Blue Sky from TSI is 800 USD in addition to a paid subscription to data services while traditional monitors cost around 25,000-30,000 USD so that is why it is called low-cost sensors. The procedure was not costly as the instrument was shipped back to the supersite lab through shipment, here we have a calibration chamber for the maintenance. It costs between 10 to 15 dollars for each for one maintenance cycle. In addition, regulatory monitoring stations require air-conditioned environments along with routine maintenance and calibration, which significantly increase operational costs. As a result, government agencies in South Asian countries often find these systems financially unfeasible. Moreover, the cost-benefit analysis point will be added to table 2 of revised manuscript (Section 3 Results & Discussion).

The updated table with estimated costs is given below:

| Country | Issues Faced | Proposed Solutions / Trouble shootings | Frequency of occurrence | *Estimated -Costs (USD/unit) |
|---|---|---|---|---|
| Sri Lanka | a) Power supply issues | a) Installing dedicated power outlets and opting solar powered backups | a) 10 | 15$ |
| | b) Network connectivity | b) using Wi-Fi dongles | b) 8 | 10$ |

| | | | |
|---|---|---|---|
| | c) clogged mesh due to salinity | c) regular maintenance | c) 4 | ~8-10$ |
| | d) firmware updates | d) manual firmware update | d) 1 | - |
| | e) Malfunctioning sensors (PM2.5 = 0 for more than 1 Day) | e) cleansing, troubleshooting and replacement | e) 1 | ~20-25$ |
| Nepal | a) Power supply disruptions | a) Coordination with local partners, and physical field visits | a) 4 | - |
| | b) Wi-Fi connectivity (Change in Wi-Fi password) | b) Opting for the solar-powered backup with independent internet sources | b) 1 | ~15-20$ |
| | c) PM2.5 Monitor defects | c) Downloading data from SD card | c) 1 | - |
| | d) Dead Power Supplies | d) Replacing malfunction adapters | d) 1 | ~10-15$ |
| | e) Firmware updates | e) manually updating firmware | e) 1 | - |
| | f) reconfiguration button issue | f) applying WD-40 spray (anti-rust) | f) 1 | - |
| Maldives | a) Continuous Wi-Fi and power supply outages | a) Using external modems, downloading data from SD cards and securing power switches | a) 14 | 10$ |
| | b) sensor maintenance due to dust and salinity | b) cleaning sensors regularly | b) 12 | - |
| | c) sensor deployment and logistic issues | c) Raising awareness and site visits for sensors installation | - | - |
| | d) Lack of reference stations for calibration | d) Calibration against regional reference stations | - | - |
| Bhutan | a) Wi-Fi connectivity | a) Verifying internet and sensor connections | a) 2 | ~15-20$ |
| | b) firmware upgrades | b) exploring firmware upgrade solutions | b) 1 | - |
| | c) data recording intervals | c) cleansing and maintenance | c) 2 | - |
| | d) Electricity outages | d) new power supply | d) 1 | ~10-15$ |
| | e) remote area access and road obstructions | e) efficient logistics for remote access and collaborating with local | e) 1 | - |

| Country | Challenge | Rectification | Count | Cost |
|---|---|---|---|---|
| | | authorities for road clearance | | |
| Bangladesh | a) Wi-Fi access (Frequent change in Wi-Fi password) | a) Using cellular modems | a) ~10-15 | 15$ |
| | b) Electricity disruptions | b) Replacing malfunctioning adapters | b) 2 | 10$ |
| | c) Sensor defects (PM2.5 = 0 for more than 1 Day) | c) Regular physical inspection and constant communication with hosts | c) 1 | 10$ |
| | d) Firmware updates | d) manufacturer coordination for manually upgradation of firmware | d) 1 | - |
| | e) Security breaches | e) Involving the public and ensuring security concerns while selecting the areas/locations to deploy the sensors | e) 1 | - |
| Pakistan | a) Bad Electronic Board | a) Replacement with locally assembled board | a) 5 | 10$ |
| | b) Sensirion Malfunctioning (PM2.5 = 0 for more than 1 Day) | b) Cleansing & replacement of component if dust clogged goes inside the sensirion. | b) 6 | 25$ |
| | c) Dead Power Supplies | c) Locally available same specifications power adapter with little modifications | c) 10 | 10$ |
| | d) Webs & Bugs inside the sensor box | d) Monthly cleaning, maintenance, and check ups | d) ~5-10 | 10$ |
| | e) Internet Disruptions | e) Wi-Fi dongles for internet provision | e) 10 | 12$ |
| | f) harsh weather conditions (reconfiguration button issues) | f) applying WD-40 (anti-rust) to setting up sensors | f) 3 | 5$ |

*The rectification costs were estimated based on the expenses reported by each collaborator. However, the costs are expected to fall within a similar range across all South Asian countries, as detailed records of expenses have not been maintained.*

4. Some content in the Introduction would be more appropriate in the Methods section. Reorganizing the manuscript to clarify methods and improve flow would make it easier for readers to follow the study's approach and findings.

**Reply:** Noted and changes will be incorporated in the revised manuscript. Authors revised the manuscript by restructuring moving headings 1.4 and heading 1.6 under Section 2 Methodology.

5. Table 2 summarizes technical challenges and solutions but does not specify whether this information is based solely on the authors' observations or if it includes input from formal surveys or consultations. If surveys were conducted, please include details on the methodology (including any ethics statement), respondent demographics, survey methods, and response rates to enhance transparency.

**Reply:** The observations summarized in Table 2 are based on the issues and challenges shared by the network operators from each country, who are also co-authors of this study. Beyond these contributions, no formal surveys or consultations were conducted. To enhance clarity, we will include a detailed description of authors' observations in the methodology section of the revised manuscript.

6. The introduction specifically and the manuscript more generally would benefit from additional context on the scientific basis for low-cost sensors, particularly around sensor accuracy, limitations, and typical calibration challenges in the field. This would provide readers with a stronger foundation for understanding the study's objectives and limitations.

**Reply:** This is a good point. There has been a great deal of work on low-cost sensors, particularly those measuring PM2.5, that has added greatly to our understanding of their accuracy and precision. For example, Zheng et al. (Atmos. Meas. Tech., 2018) found that with proper calibration the accuracy of low-cost PM2.5 sensors in both high and low concentration environments can be ~10%. In addition, there are community wide approaches to calibration and quality control that have greatly increased our ability to manage and maintain low-cost sensor networks to ensure accuracy and precision (Barkjohn et al., EST Air, 2024). Furthermore, the introduction of the manuscript will be revised to address objectives and limitations of the study.

**References:**

Barkjohn, K.K., Clements, A., Mocka, C., Barrette, C., Bittner, A., Champion, W., Gantt, B., Good, E., Holder, A., Hillis, B. and Landis, M.S., 2024. Air Quality Sensor Experts Convene: Current Quality Assurance Considerations for Credible Data. *ACS ES&T Air*, *1*(10), pp.1203-1214.

Khaleel, N., Schauer, J.J., Bergin, M.H., Jani, S.J.M., Bhave, P.V., Razzaq, T.A. and Khan, M.F., 2024. Differentiating local and regional drivers of exceedances of WHO PM2. 5 standards with a low-cost sensor network in the greater male'region (GMR). *Atmospheric Pollution Research*, p.102341.

Madhwal, S., Tripathi, S. N., Bergin, M. H., Bhave, P., de Foy, B., Reddy, T. R., Chaudhry, S. K., Jain, V., Garg, N., & Lalwani, P. (2024). Evaluation of PM2.5 spatio-temporal variability and hotspot formation using low-cost sensors across urban-rural landscape in Lucknow, India. *Atmospheric Environment*, *319*, 120302.

Senarathna, M. et al. Enhancing Low-Cost PM2.5 Air Quality Monitoring Sensors through Sensor Calibration Using Linear Regression and Echo-State Network Models (*in preparation*).

Shabbir, M., Saeed, T., Saleem, A., Bergin, M. H., Bhave, P., & Khokhar, Md. F. A Paradigm Shift: Low-Cost Sensors for Effective Air Quality Monitoring and Management in Developing Countries. *Environmental Technology & Innovation* (in review).

Shreshta, H., Sapkota, R. P., Bhave, P. V., Bergin, M. H., Adhikary, B., Mool, E., Pokhrel, S., Bajgain, T. R., & Maskey Byanju, R. Field calibration and performance of low-cost sensors in Kathmandu Valley for ambient PM2.5 monitoring. *International Journal of Environmental Research Journal of Environmental Management* (in review).

Zaman, S.U. et al. Quantifying Transboundary Air Pollution in Bangladesh: Seasonal Trends and Variations in PM2.5 Concentrations Using Low-cost Sensor Network (*in preparation*).

Zheng, T., Bergin, M.H., Johnson, K.K., Tripathi, S.N., Shirodkar, S., Landis, M.S., Sutaria, R. and Carlson, D.E., 2018. Field evaluation of low-cost particulate matter sensors in high-and low-concentration environments. *Atmospheric Measurement Techniques, 11*(8), pp.4823-4846.

---

## Author Comment (AC2)

**Reply to the comments of Referee # 2**

**Preprint Details:**

**Manuscript number:** egusphere-2024-1932

**Title:** Sustaining Low-Cost PM$_{2.5}$ Monitoring Networks in South Asia: Technical Challenges and Solutions

**Authors:** Talha Saeed, Muhammad Mahad Khaliq, Michael Howard Bergin, Prakash V. Bhave, Noora Khaleel, Enna Mool, Mahesh Senarathna, Shahid Uz Zaman, Shatabdi Roy, Abdus Salam, Jas Raj Subba, Muhammad Fahim Khokhar*

**Authors Acknowledgment:**

The authors sincerely thank the referees for their thorough reviews and insightful comments. We have carefully addressed each comment by acknowledging the constructive feedback and incorporating it into the revised manuscript to enhance its quality and clarity.

**Referee # 2 Comments:**

The authors are appreciated for their efforts in maintaining an extensive PM sensor network comprising 380 TSI BlueSky sensors across various South Asian countries (excluding India and Afghanistan). This manuscript represents a commendable attempt to outline various troubleshooting methods for sustaining the TSI BlueSky sensor network over extended periods. However, a few concerns need to be addressed to improve the clarity and robustness of the manuscript.

**Specific Comments:**

1. The manuscript describes a study across South Asia but does not include data from India. This raises questions, as sensors in India are marked in Figure 1. It would be good to specify why these countries are not part of the study despite phrasing as a study in "South Asia" multiple times in the manuscript.

**Reply:** That's valid point and we suggest there should be a change in title replacing South Asia with "South Asian countries". Some of the sensors are solely for personal use so their display depends on their user control setting whether they want to show publicly sensors or not. While the Indian team didn't opt for it, the sensors displayed on the TSI dashboard are of some users (not part of this project) who bought the sensors for their personal usage and made them available publicly.

2. The description of the calibration and collocation protocols used in the study lacks sufficient detail. Please elaborate on the procedures followed to ensure data reliability and accuracy, as that is a major concern when incorporating the LCS network into the regulatory framework. It is not demonstrated that calibration before and after cleaning was consistent across all countries in the study. Addressing this gap would strengthen the reliability of the results.

**Reply:** Calibration and collocation are described in other companion manuscripts Jain et al (2024); Madhwal et al. (2024); Khaleel et al. (2024); Shabbir et al. (in review); Shrestha et al. (in review); Zaman et al. (in preparation); Senarathna et al. (in review). After working for 3 consistent years these sensors due to exposure of dust, needed calibration, so Pakistan team made a calibration chamber (patent submitted) in which first they calibrated the sensors in batches and computed the offset of each individual sensor (See Section 1.6.3 Sensor Harmonization which will be moved to Section 2 Methodology in revised manuscript to address clarity issues). Calibration procedures before and after cleaning varied among all countries, with the Pakistan team exhibiting a more consistent and rigorous approach to regular cleaning and calibration protocols. Moreover, troubleshooting workshop on 24 May 2023 was conducted for all partner countries including India, in which they were trained for

issues like zero error due to dust clogging. Another important factor to consider is that each country experiences different weather and environmental conditions. For instance, Bhutan and Maldives have relatively low levels of pollution, resulting in fewer technical challenges but greater logistical challenges compared to other countries such as Nepal, Bangladesh, Sri Lanka, and Pakistan. However, it was recommended for all teams to go for cleaning and calibration cycle every year.

3. The introduction and methodology sections need significant restructuring for clarity. A considerable portion of the methodology is currently included in the introduction, which should focus on background and objectives. Does "Deployment" in Table 1 refer to the starting month of sensor deployment? Additionally, for how many months was the network operational? Revising these sections will enhance the manuscript's readability.

**Reply:** Authors will be revised the manuscript by restructuring Section 1.4 and Section 1.6 to Section 2 Methodology. Yes, the deployment refers to the month in which the sensor was on field for monitoring $PM_{2.5}$. The whole network has been sustained and maintained for about 3 years now and these sensors are still deployed and collecting data from mentioned countries (except India).

4. The authors mention dust accumulation and environmental exposure but fail to specify the measurement period or the specific environmental conditions the sensors were exposed to. Additionally, how often were the described failures observed across the network? Detailed insights into failure frequency would strengthen the analysis.

**Reply:** That's a good question. The measurement period was from the date of deployment and most of the sensors were exposed to heavy traffic pollution loads. The sensors were afterwards cleaned & calibrated, and harmonization factors computed before and after of which the results are shown in table 3.

The frequency of such failures varies depending on the surrounding environmental conditions or specific seasons. For instance, sensor NUST-Pk4 has clogged inlet due to construction activity in its vicinity (as shown in Figure 10). Similar cases have been reported from the network operators of other partner countries except Bhutan and Maldives.

The updated table with frequencies of occurrence, downtime and costs is given below:

| Country | Issues Faced | Proposed Solutions / Trouble shootings | Frequency of occurrence | *Estimated-Costs (USD/unit) |
|---|---|---|---|---|
| Sri Lanka | a) Power supply issues | a) Installing dedicated power outlets and opting solar powered backups | a) 10 | 15$ |
| | b) Network connectivity | b) using Wi-Fi dongles | b) 8 | 10$ |
| | c) clogged mesh due to salinity | c) regular maintenance | c) 4 | ~8-10$ |
| | d) firmware updates | d) manual firmware update | d) 1 | - |
| | e) Malfunctioning sensors (PM2.5 = 0 for more than 1 Day) | e) cleansing, troubleshooting and replacement | e) 1 | ~20-25$ |
| Nepal | a) Power supply disruptions | a) Coordination with local partners, and physical field visits | a) 4 | - |
| | b) Wi-Fi connectivity (Change in Wi-Fi password) | b) Opting for the solar-powered backup with independent internet sources | b) 1 | ~15-20$ |
| | c) PM2.5 Monitor defects | c) Downloading data from SD card | c) 1 | - |
| | d) Dead Power Supplies | d) Replacing malfunction adapters | d) 1 | ~10-15$ |
| | e) Firmware updates | e) manually updating firmware | e) 1 | - |
| | f) reconfiguration button issue | f) applying WD-40 spray (anti-rust) | f) 1 | - |
| Maldives | a) Continuous Wi-Fi and power supply outages | a) Using external modems, downloading data from SD cards and securing power switches | a) 14 | 10$ |
| | b) sensor maintenance due to dust and salinity | b) cleaning sensors regularly | b) 12 | - |
| | c) sensor deployment and logistic issues | c) Raising awareness and site visits for sensors installation | - | - |
| | d) Lack of reference stations for calibration | d) Calibration against regional reference stations | - | - |
| Bhutan | a) Wi-Fi connectivity | a) Verifying internet and sensor connections | a) 2 | ~15-20$ |

| | | | | |
|---|---|---|---|---|
| | b) firmware upgrades | b) exploring firmware upgrade solutions | b) 1 | - |
| | c) data recording intervals | c) cleansing and maintenance | c) 2 | - |
| | d) Electricity outages | d) new power supply | d) 1 | ~10-15$ |
| | e) remote area access and road obstructions | e) efficient logistics for remote access and collaborating with local authorities for road clearance | e) 1 | - |
| Bangladesh | a) Wi-Fi access (Frequent change in Wi-Fi password) | a) Using cellular modems | a) ~10-15 | 15$ |
| | b) Electricity disruptions | b) Replacing malfunctioning adapters | b) 2 | 10$ |
| | c) Sensor defects (PM2.5 = 0 for more than 1 Day) | c) Regular physical inspection and constant communication with hosts | c) 1 | 10$ |
| | d) Firmware updates | d) manufacturer coordination for manually upgradation of firmware | d) 1 | - |
| | e) Security breaches | e) Involving the public and ensuring security concerns while selecting the areas/locations to deploy the sensors | e) 1 | - |
| Pakistan | a) Bad Electronic Board | a) Replacement with locally assembled board | a) 5 | 10$ |
| | b) Sensirion Malfunctioning (PM2.5 = 0 for more than 1 Day) | b) Cleansing & replacement of component if dust clogged goes inside the sensirion. | b) 6 | 25$ |
| | c) Dead Power Supplies | c) Locally available same specifications power adapter with little modifications | c) 10 | 10$ |
| | d) Webs & Bugs inside the sensor box | d) Monthly cleaning, maintenance, and check ups | d) ~5-10 | 10$ |
| | e) Internet Disruptions | e) Wi-Fi dongles for internet provision | e) 10 | 12$ |
| | f) harsh weather conditions (reconfiguration button issues) | f) applying WD-40 (anti-rust) to setting up sensors | f) 3 | 5$ |

*The rectification costs were estimated based on the expenses reported by each collaborator. However, the costs are expected to fall within a similar range across all South Asian countries, as detailed records of expenses have not been maintained.*

5. The authors state that the troubleshooting and repair methodology improved the network's reliability and accuracy in Pakistan. Since the manuscript is pointing to the study in South Asia, the readers will expect results on similar benefits observed in other countries.

**Reply:** That's a valid point and thank you for raising this important point. The sensors were cleaned and harmonized in other countries as well, and the results from other countries will be added into the revised manuscript (Table 2). The authors are confident that implementing the cleaning routines, calibration, and validation protocols outlined in this manuscript can enhance the lifespan and data quality of sensors globally. However, we expect that the frequency of cleaning and calibration may vary depending on environmental conditions, with longer intervals likely in cleaner regions such as the Maldives or Bhutan. To ensure consistency, a regional troubleshooting workshop was organized by the Pakistan team, with participation from all partner countries, including India. During the workshop, all teams were trained in the cleaning and calibration procedures described in this manuscript. Additionally, the South Asian countries regularly coordinate efforts to sustain the low-cost sensor network and address these challenges.

6. To enhance clarity in Table 2, consider presenting the issues faced alongside their corresponding solutions or troubleshooting methods rather than listing all solutions and issues separately. Also, it would be good to add the frequency of the issues in each country. As the manuscript only deals with one brand of PM sensor, a detailed explanation of the issues, the cost involved in the troubleshooting, the downtime faced by each issue, etc. can be useful for the readers.

**Reply:** That's a valid point to enhance readability and clarity. The details will be added into table 2 in the revised manuscript as mentioned above.

7. While heat waves are mentioned as a factor accelerating corrosion, there is insufficient detail on the frequency of corrosion issues across the 380 PM sensors. Are these findings specific to a particular site, or do they represent regional trends? Additionally, the evaluation of solar panels is unclear.

**Reply:** The observation regarding heat waves as a potential factor in accelerating corrosion is valid, as this aspect is not always explicitly considered in sensor evaluations. However, it is important to note that electronic malfunctions of sensors were predominantly observed during the summer and monsoon seasons. This temporal correlation highlights the potential

impact of environmental factors, including extreme weather conditions on sensor performance (also reported by Holstius et al., 2014; Kelly et al., 2017; Jayaratne et al., 2018).

We have modified the lines 265 to 285 in section 3.3 as following to address:

Air quality sensors are critical components in monitoring environmental pollution and providing data for public health advisories. However, these instruments are susceptible to environmental factors that can impair their functionality. In Islamabad, weather extremes, such as heat waves, high humid conditions especially during monsoon & winter smog episodes in addition to power outages present significant challenges to the operational integrity of these sensors. This part examines the correlation between harsh weather conditions, specifically heat waves, and the likelihood of sensor malfunctions due to rusting of pins, decreased power output, and power supply failures. The temperature in Figure 11 indicates that Islamabad experiences high temperatures, particularly from May to August, with peaks that could potentially exceed 40°C. Alongside, the relative humidity levels during these months decrease, indicating the occurrence of heat waves. Such conditions are known to accelerate the corrosion process, causing rusting of sensor pins. The precipitation chart shows substantial rainfall in July and August. While precipitation may not directly cause rusting, the resulting increase in humidity levels after rainfall can create conditions conducive to corrosion, especially if sensors are not properly maintained. Especially, rusting incidents of sensor parts are reported by partner countries like Sri Lanka and Maldives. Wind can act as a catalyst for heat dissipation, which might help in cooling down the sensors during heat waves. However, it can also introduce particulate matter that could accumulate on sensors, potentially leading to clogging of sensor fans leading to zero $PM_{2.5}$ values. Furthermore, soaring temperatures during heat waves can lead to electronic board and power supply failures (Table 2).

**Solar Explanation**

To address the specific operational challenges (e.g. power outages), the BlueSky sensors, originally designed for electricity-based (5volts~5watts) operations, were modified by the Pakistan team to also function using solar-powered backup. This modification was undertaken to extend the applicability of Bluesky sensors to regions lacking reliable electricity and Wi-Fi connectivity. The modifications aimed to enhance the resilience and versatility of the sensors in diverse operational environments, thereby increasing their suitability for deployment in remote or resource-constrained areas.

Following lines will be added under Section 3.4 of the revised manuscript:

Table 4 depicts an experiment while the BlueSky sensor was powered using a 12V, 26Ah battery (referred to as Trial 1), which provided a runtime of approximately 422 hours (17.5

days) under standalone operation. Subsequently, a 30W, 18V solar panel was integrated into the system to simultaneously power the BlueSky sensor and to recharge the battery. Under these conditions, the system operated for approximately 74 hours (Trial 2), supporting both the sensor's operations and the concurrent charging of the battery suggesting that it could be an alternative source of power supply in remote areas and/or special conditions such as power outages.

**Technical corrections:**

1. Ensure that all figures and tables are appropriately cited in the manuscript.
   **Reply:** The complete manuscript has been revised and ensured that all figures and tables are properly cited in the text.
2. Avoid repetition of sentences.
   **Reply:** The manuscript has gone through extensive revision by all the authors to avoid repeated sentences.

**References:**

Holstius, D. M., Pillarisetti, A., Smith, K. R., and Seto, E.: Field calibrations of a low-cost aerosol sensor at a regulatory monitoring site in California, Atmos. Meas. Tech., 7, 1121–1131, https://doi.org/10.5194/amt-7-1121-2014

Jain, V., Mukherjee, A., Banerjee, S., Madhwal, S., Bergin, M.H., Bhave, P., Carlson, D., Jiang, Z., Zheng, T., Rai, P. and Tripathi, S.N., 2024. A hybrid approach for integrating micro-satellite images and sensors network-based ground measurements using deep learning for high-resolution prediction of fine particulate matter (PM2. 5) over an Indian city, Lucknow. *Atmospheric Environment*, *338*, p.120798.

Jayaratne, R., Liu, X., Thai, P., Dunbabin, M., and Morawska, L.: The influence of humidity on the performance of a low-cost air particle mass sensor and the effect of atmospheric fog, Atmos. Meas. Tech., 11, 4883–4890, https://doi.org/10.5194/amt-11-4883-2018

Kelly, K. E., Whitaker, J., Petty, A., Widmer, C., Dybwad, A., Sleeth, D., Martin, R., and Butterfield, A.: Ambient and laboratory evaluation of a low-cost particulate matter sensor, Environ. Pollut., 221, 491–500, 2017

Khaleel, N., Schauer, J.J., Bergin, M.H., Jani, S.J.M., Bhave, P.V., Razzaq, T.A. and Khan, M.F., 2024. Differentiating local and regional drivers of exceedances of WHO PM2. 5 standards with a low-cost sensor network in the greater male'region (GMR). *Atmospheric Pollution Research*, p.102341.

Madhwal, S., Tripathi, S. N., Bergin, M. H., Bhave, P., de Foy, B., Reddy, T. R., Chaudhry, S. K., Jain, V., Garg, N., & Lalwani, P. (2024). Evaluation of $PM_{2.5}$ spatio-temporal variability and

hotspot formation using low-cost sensors across urban-rural landscape in Lucknow, India. *Atmospheric Environment*, *319*, 120302.

Senarathna, M. et al. Enhancing Low-Cost PM$_{2.5}$ Air Quality Monitoring Sensors through Sensor Calibration Using Linear Regression and Echo-State Network Models (*in preparation*).

Shabbir, M., Saeed, T., Saleem, A., Bergin, M. H., Bhave, P., & Khokhar, Md. F. A Paradigm Shift: Low-Cost Sensors for Effective Air Quality Monitoring and Management in Developing Countries. *Environmental Technology & Innovation* (in review).

Shreshta, H., Sapkota, R. P., Bhave, P. V., Bergin, M. H., Adhikary, B., Mool, E., Pokhrel, S., Bajgain, T. R., & Maskey Byanju, R. Field calibration and performance of low-cost sensors in Kathmandu Valley for ambient PM$_{2.5}$ monitoring. *International Journal of Environmental Research Journal of Environmental Management* (in review).

Zaman, S.U. et al. Quantifying Transboundary Air Pollution in Bangladesh: Seasonal Trends and Variations in PM$_{2.5}$ Concentrations Using Low-cost Sensor Network (*in preparation*).